# Auditing Fairness by Betting

**Ben Chugg[1], Santiago Cortes-Gomez[1], Bryan Wilder[1], Aaditya Ramdas[1,2]**
Departments of Machine Learning[1] and Statistics[2]
Carnegie Mellon University
{benchugg, scortesg, bwilder, aramdas}@cmu.edu

## Abstract

We provide practical, efficient, and nonparametric methods for auditing the fairness of deployed classification and regression models. Whereas previous work relies on a fixed-sample size, our methods are sequential and allow for the continuous monitoring of incoming data, making them highly amenable to tracking the fairness of real-world systems. We also allow the data to be collected by a probabilistic policy as opposed to sampled uniformly from the population. This enables auditing to be conducted on data gathered for another purpose. Moreover, this policy may change over time and different policies may be used on different subpopulations. Finally, our methods can handle distribution shift resulting from either changes to the model or changes in the underlying population. Our approach is based on recent progress in anytime-valid inference and game-theoretic statistics—the "testing by betting" framework in particular. These connections ensure that our methods are interpretable, fast, and easy to implement. We demonstrate the efficacy of our approach on three benchmark fairness datasets.

## 1 Introduction

As algorithmic decision-making continues to increase in prevalence across both the private and public sectors [1, 2], there has been an increasing push to scrutinize the fairness of these systems. This has lead to an explosion of interest in so-called "algorithmic fairness", and a significant body of work has focused on both defining fairness and training models in fair ways (e.g., [3–5]). However, preventing and redressing harms in real systems also requires the ability to *audit* models in order to assess their impact; such algorithmic audits are an increasing area of focus for researchers and practitioners [6–9]. Auditing may begin during model development [10], but as model behavior often changes over time throughout real-world deployment in response to distribution shift or model updates [11, 12], it is often necessary to repeatedly audit the performance of algorithmic systems over time [8, 13]. Detecting whether deployed models continue to meet various fairness criteria is of paramount importance to deciding whether an automated decision-making system continues to act reliably and whether intervention is necessary.

In this work, we consider the perspective of an auditor or auditing agency tasked with determining if a model deployed "in the wild" is fair or not. Data concerning the system's decisions are gathered over time (perhaps with the purpose of testing fairness but perhaps for another purpose) and our goal is to determine if there is sufficient evidence to conclude that the system is unfair. If a system is in fact unfair, we want to determine so as early as possible, both in order to avert harms to users and because auditing may require expensive investment to collect or label samples [8, 13]. Following the recent work of Taskesen et al. [14] and Si et al. [15], a natural statistical framework for thinking about this problem is hypothesis testing. Informally, consider the null and alternative hypotheses, $H_0$ and $H_1$, defined respectively as

$$H_0 : \text{the model is fair}, \quad H_1 : \text{the model is unfair}.$$

37th Conference on Neural Information Processing Systems (NeurIPS 2023).

Unfortunately, traditional hypothesis testing requires stringent assumptions on the data; a fixed number of iid data points, for example. Such assumptions are unrealistic in our setting. We should, for instance, be able to continually test a system as we receive more information, i.e., perform *sequential* hypothesis testing. Moreover, we would like to be able stop collecting additional samples at arbitrary data-dependent stopping times if we have sufficient evidence against the null. This is not allowed in traditional statistical frameworks, where it is known as "peeking" or "p-hacking."

To overcome these challenges, we take advantage of recent progress in safe, anytime-valid inference (SAVI) to construct sequential procedures for determining whether an existing decision-making system is fair. SAVI is part of *sequential analysis*, a branch of statistics concerned—as the name suggests—with analyzing data sequentially while maintaining statistical validity. This subfield traces its origins back to Wald, Lai, Robbins, and several others beginning in the 1940s [16–20]. More recently, the power of sequential analysis to enable inference under continuous monitoring of data and data-dependent stopping rules (hence *safe* and *anytime-valid*) has led to an explosion of work in the area [21–23]. A further exciting development has been the connection between such methods and so-called "game-theoretic probability" [24, 25] which is consistent with modern measure theoretic probability[1] but provides an alternative foundation based on repeated games. Importantly for our purposes, this renders many of the tools of SAVI interpretable as well as statistically powerful. We refer the interested reader to the recent survey by Ramdas et al. [27] for further detail on the relationship between sequential analysis, SAVI, and game-theoretic statistics.

**Contributions.** We develop tools to sequentially audit both classifiers and regressors. In particular:

1. We formulate the problem of auditing the fairness of classification and regression models in terms of sequential hypothesis testing. The focus on *sequential* testing is distinct from other work, and highlights various desiderata that are important in practice—specifically, (i) being able to continuously monitor the data and (ii) a focus on rejecting the null as early as possible (i.e., reducing the number of costly samples needed to detect unfairness).

2. Next, we design nonparametric sequential hypothesis tests which hold under various definitions of group fairness. We treat auditing as sequential two-sample testing and adapt the recent work of Shekhar and Ramdas [28] on two-sample testing by betting to the fairness setting. We also provide novel bounds on the expected stopping time of our tests under the alternative, and demonstrate how to handle distribution drift (due to either model changes or changes in the underlying population), time-varying data collection policies, and composite nulls which more accurately reflect practical demands.

3. Finally, we demonstrate the real world applicability of our methods on three datasets: credit default data, US census data, and insurance data. We show that our method is robust to distribution shift resulting from model retraining and to various randomized data-collection policies whose densities deviate significantly from the underlying population. All code is publicly available at https://github.com/bchugg/auditing-fairness.

In order to provide a preview of our methodology, we suggest the following thought experiment. Imagine a fictitious better who is skeptical that a machine learning system is fair. She sets up an iterated game wherein she bets on the results of the audit as it is conducted. Her bets are structured such that, if the system is unfair, her expected payoff will be large. Conversely, if the system is fair, her expected payoff is small. Thus, if her wealth increases over time, it is evidence that the system is unfair. Our sequential test thus rejects the null if her wealth crosses some predetermined threshold. Of course, the trick is to design her bets in such a way the above conditions are satisfied, and that under the alternative, her wealth grows as quickly as possible. Mathematically, the above is made rigorous via nonnegative (super)martingales and results concerning their behavior over time [29, 30].

**Additional Related Work.** This work sits at the intersection of several fields. On the fairness side, the closest work to ours is that of Taskesen et al. [14] and Si et al. [15], both of which study fairness through the lens of (fixed-time) hypothesis testing, based on a batch of $n$ iid observations. Given iid data $Z_t \sim \rho$, they formulate the null as $H_0 : \rho \in \mathcal{G}$, where $\mathcal{G}$ is the set of all "fair" distributions and derive a test statistic based on projecting the empirical distribution onto $\mathcal{G}$ (minimizing a Wasserstein distance). They work with classifiers only. On the more technical side, we view our work as

---

[1]There are some subtleties to this remark, but they are outside the scope of this work. However, as an example of how the insights of game-theoretic statistics can be applied to modern probability, we highlight the recent work of Waudby-Smith and Ramdas [26] which draws inspiration from Shafer and Vovk [25].

a continuation and application of the testing by betting framework [21] and more specifically of nonparametric two sample testing by betting [28]. We will deploy similar "betting strategies" (to be defined later) as [28], but provide novel analyses which are particular to our setting, and consider several extensions. Relatedly, but slightly further removed from our setting, Duan et al. [31] also employ game-theoretic ideas to construct interactive rank tests. The idea of a auditing a system to verify its veracity extends well beyond the domain of algorithmic fairness. Statistical procedures have been developed, for instance, to audit election results [32] and recently several SAVI inspired methods have been developed for several such scenarios [33, 34]. More broadly and outside the scope of this paper, betting techniques have been deployed with great success in optimization, online learning, and statistics [35–38, 26].

## 2   Preliminaries

Consider a feature space $\mathcal{X}$ and model $\varphi : \mathcal{X} \to [0, 1]$. Depending on the application, $\varphi(x)$ might be a risk score for an individual with covariates $x$, a classification decision, or the probability that some action is taken. Each $x \in \mathcal{X}$ is associated with some "sensitive attribute" $A = A_x \in \{0, 1, \dots, J\}$ (indicating, e.g., the result of a private health test, whether they went to college, their income strata, etc). We will often refer to $A$ as group membership. The classifier may or may not observe $A$. For the sake of exposition, we will assume that $A \in \{0, 1\}$, i.e, there are two groups. However, our methods extend straightforwardly to more than two groups. The details are provided in Appendix B. Before formally stating the problem, let us discuss the notion of fairness we will employ in this work.

**Fairness.** We focus on the concept of "group" fairness. Roughly speaking, this involves ensuring that groups of individuals sharing various attributes are treated similarly. There have been multiple notions of group fairness proposed in the literature. We introduce the following generalized notion of group fairness, which can be instantiated to recapture various others.

**Definition 1.** Let $\{\xi_j(A, X, Y)\}_{j \in \{0, \dots, J\}}$ denote a family of conditions on the attributes $A$, covariates $X$, and outcomes $Y$. We say a predictive model $\varphi : \mathcal{X} \to [0, 1]$ is fair with respect to $\{\xi_j\}$ and a distribution $\rho$ over $\mathcal{X}$ if, for all $i, j \in [J]$, $\mathbb{E}_{X \sim \rho}[\varphi(X)|\xi_i(A, X, Y)] = \mathbb{E}_{X \sim \rho}[\varphi(X)|\xi_j(A, X, Y)]$.

As was mentioned above, unless otherwise stated we will assume that there are only two conditions $\xi_0, \xi_1$. For our purposes, the important feature of Definition 1 is that it posits the equality of means. Indeed, letting $\mu_b = \mathbb{E}_{X \sim \rho}[\varphi(X)|\xi_b]$ for $b \in \{0, 1\}$ we can write our null as $H_0 : \mu_0 = \mu_1$ and alternative as $H_1 : \mu_0 \neq \mu_1$. Different choices of conditions $\xi_j$ lead to various fairness notions in the literature. For instance, if $\varphi$ is a classification model, then:

1. Taking $\xi_0 = \{A = 0, Y = 1\}$ and $\xi_1 = \{A = 1, Y = 1\}$ results in *equality of opportunity* [39]. *Predictive equality* [40] is similar, corresponding to switching $Y = 1$ with $Y = 0$.

2. Taking $\xi_0 = \{A = 0\}, \xi_1 = \{A = 1\}$ results in *statistical parity* [41].

3. Taking $\xi_j = \{A = j, \ell(X)\}$ for some projection mapping $\ell : \mathcal{X} \to F \subset \mathcal{X}$ onto "legitimate factors" results in *conditional statistical parity* [42, 41].

**Problem Formulation.** We consider an *auditor* who is receiving two streams of predictions $Z^0 = (\varphi(X_t^0))_{t \in T_0}$ and $Z^1 = (\varphi(X_t^1))_{t \in T_1}$, where $X_t^b$ obeys condition $\xi_b$ (i.e., is drawn from some distribution over $\mathcal{X}|\xi_b$). For brevity, we will let $\widehat{Y}_t^b = \varphi(X_t^b)$, $b \in \{0, 1\}$. The index sets $T_0, T_1 \subset \mathbb{N} \cup \{\infty\}$ denote the times at which the predictions are received. We let the index differ between groups as it may not be feasible to receive a prediction from each group each timestep. We refer to this entire process as an *audit* of the model $\varphi$.

For $b \in \{0, 1\}$, let $T_b[t] = T_b \cap [t]$ be the set of times at which we receive predictions from group $b$ up until time $t$. The auditor is tasked with constructing a *sequential hypothesis test* $\phi \equiv (\phi_t)_{t \geq 1}$ where $\phi_t = \phi_t((\cup_{t \in T_0[t]} Z_t^0) \cup (\cup_{t \in T_1[t]} Z_t^1)) \in \{0, 1\}$ is a function of all predictions received until time $t$. We interpret $\phi_t = 1$ as "reject $H_0$", and $\phi_t = 0$ as "fail to reject $H_0$." Once we reject the null, we stop gathering data. That is, our stopping time is $\tau = \arg \inf_t \{\phi_t = 1\}$. We say that $\phi$ is a *level-$\alpha$* sequential test if

$$\sup_{P \in H_0} P(\exists t \geq 1 : \phi_t = 1) \leq \alpha, \quad \text{or equivalently} \quad \sup_{P \in H_0} P(\tau < \infty) \leq \alpha. \tag{1}$$

In words, the test should have small false positive rate (type I error) *simultaneously across all time steps*. (Note that a fixed-time level-$\alpha$ test would simply drop the quantifier $\exists t \geq 1$ and concern

itself with some fixed time $t = n$.) We also wish to design tests with high power. Formally, we say that $\phi$ has *asymptotic power* $1 - \beta$ if $\sup_{P \in H_1} P(\forall t \geq 1 : \phi_t = 0) \leq \beta$, or equivalently $\sup_{P \in H_1} P(\tau = \infty) \leq \beta$. That is, in all worlds in which the alternative is true, we fail to reject with probability at most $\beta$ (type II error). Typically the type-II error $\beta \equiv \beta_n$ decreases with the sample size $n$. In this work we will develop asymptotic power one tests, meaning that $\beta_n \to 0$ as $n \to \infty$.

**Martingales and filtrations.** Our techniques rely on the machinery of nonnegative (super)martingales and results concerning their behavior over time. We introduce some of the technicalities here. A (forward) filtration $\mathcal{F} \equiv (\mathcal{F}_t)_{t \geq 0}$ is an increasing sequence of $\sigma$-fields $\mathcal{F}_t \subset \mathcal{F}_{t+1}$. Throughout this paper, we will consider the "canonical filtration" $\mathcal{F}_t = \sigma(Z_1, \ldots, Z_t)$ which can heuristically (but very usefully) be thought of as all the information known at time $t$. We say that a stochastic process $S = (S_t)_{t \geq 1}$ is *adapted* to $(\mathcal{F}_t)$ if $S_t$ is $\mathcal{F}_t$ measurable for all $t \geq 1$, and *predictable* if $S_t$ is $\mathcal{F}_{t-1}$ measurable for all $t \geq 1$. A *P-martingale* is an adapted stochastic process $M = (M_t)_{t \geq 1}$ such that $\mathbb{E}_P[M_{t+1}|\mathcal{F}_t] = M_t$ for all $t \geq 1$. If the equality is replaced with $\leq$, then $M$ is a *P-supermartingale*. A particularly useful result in sequential analysis is *Ville's inequality* [29], which states that if $M$ is a nonnegative $P$-supermartingale, then for all $\alpha > 0$, $P(\exists t \geq 0 : M_t \geq 1/\alpha) \leq \alpha \mathbb{E}_P[M_0]$. We will also employ the following randomized improvement to Ville's inequality [43, Corollary 4.1.1]: For $M$ as above and any $\mathcal{F}$-stopping time $\tau$, $P(\exists t \leq \tau : M_t \geq 1/\alpha$ or $M_\tau \geq U/\alpha) \leq \alpha$, where $U$ is a uniform random variable on $[0, 1]$ *which is independent of $M$ and $\tau$*.

## 3 Methods

We begin by introducing the methodology in the setting where the predictions are received uniformly at random from the population. We will then progressively generalize the setting: Sections 3.2 and 3.3 will enable time-varying data collection policies, Section 3.4 will allow the means $\mu_0$ and $\mu_1$ to change with time, and Section 3.5 will consider composite nulls of the form $|\mu_0 - \mu_1| \leq \epsilon$.

To ease the presentation, let us make the assumption that we receive an audit from both groups each timestep, so that $T_0 = T_1 = \mathbb{N}$. This is without loss of generality; one can simply wait until multiple audits from each group are available. However, Appendix A provides a more detailed discussion on how to modify our framework and theorems if this condition is unmet.

### 3.1 Testing by betting

Returning to the intuition for a moment, recall the fictitious bettor discussed in the introduction. She attempts to prove that the system is unfair by betting on the results of the audits before they are revealed. If the system *is* unfair, then the average difference between $\widehat{Y}_t^0$ and $\widehat{Y}_t^1$ (i.e., the model's outputs across different groups) will be non-zero. The skeptic's bets are therefore structured such that if $\mu_0 = \mathbb{E}\widehat{Y}_t^0 \neq \mathbb{E}\widehat{Y}_t^1 = \mu_1$, then her expected wealth will increase over time. More formally, at time $t$, the skeptic designs a *payoff function* $S_t : \mathbb{R} \times \mathbb{R} \to \mathbb{R}_{\geq 0}$ which is $\mathcal{F}_{t-1}$ measurable and $\mathbb{E}_P[S_t(\widehat{Y}_t^0, \widehat{Y}_t^1)|\mathcal{F}_{t-1}] \leq 1$ if $P \in H_0$ (i.e, $\mu_0 = \mu_1$). Next, the skeptic receives the model predictions $\widehat{Y}_t^0 = \varphi(X_t^0)$ and $\widehat{Y}_t^1 = \varphi(X_t^1)$. We assume the skeptic starts with wealth of $\mathcal{K}_0 = 1$. At each time $t$, she reinvests all her wealth $\mathcal{K}_{t-1}$ on the outcome and her payoff is $\mathcal{K}_t = S_t(\widehat{Y}_t^0, \widehat{Y}_t^1) \cdot \mathcal{K}_{t-1} = \prod_{i=1}^t S_t(\widehat{Y}_t^0, \widehat{Y}_t^1)$. We call $(\mathcal{K}_t)_{t \geq 0}$ the skeptic's *wealth process*. The wealth process is a supermartingale starting at 1 under the null, due to the constraint on the payoff function. Thus, by Ville's inequality, the probability that $\mathcal{K}_t$ ever exceeds $1/\alpha$ is at most $\alpha$ when the model is fair. The skeptic's goal, as it were, is to design payoff functions such that the wealth process grows quickly under the alternative, and thus rejection occurs sooner rather than later.

Inspired by a common idea in game-theoretic statistics (cf. [21, 26–28]), let us consider the following payoff function:

$$S_t(\widehat{Y}_t^0, \widehat{Y}_t^1) = 1 + \lambda_t(\widehat{Y}_t^0 - \widehat{Y}_t^1), \tag{2}$$

where $\lambda_t$ is predictable and lies in $[-1, 1]$ to ensure that $S_t(\widehat{Y}_t^0, \widehat{Y}_t^1) \geq 0$. Note that for $P \in H_0$, $\mathbb{E}_P[\mathcal{K}_t|\mathcal{F}_{t-1}] = \mathcal{K}_{t-1}\mathbb{E}_P[1 + \lambda_t(\widehat{Y}_t^0 - \widehat{Y}_t^1)|\mathcal{F}_{t-1}] = \mathcal{K}_{t-1}$, so $(\mathcal{K}_t)_{t \geq 1}$ is a nonnegative $P$-martingale. Rejecting when $\mathcal{K}_t > 1/\alpha$ thus gives to a valid level-$\alpha$ sequential test as described above. We will select $\lambda_t$ using Online Newton Step (ONS) [44, 45], which ensures exponential growth of the wealth process under the alternative. Figure 1 illustrates the behavior of the wealth process under various hypotheses when using ONS to choose $\lambda_t$. As the difference between the means $\Delta = |\mu_0 - \mu_1|$

**Algorithm 1** Testing group fairness by betting

**Input:** $\alpha \in (0,1)$
$\mathcal{K}_0 \leftarrow 1$
**for** $t = 1, 2, \ldots, \tau$ **do**
    # $\tau$ *may not be known in advance*
    Perhaps receive audits $\widehat{Y}_t^0$ and $\widehat{Y}_t^1$
    Construct payoff $S_t$ (e.g., (2), (16), or (7))
    Update $\mathcal{K}_t \leftarrow \mathcal{K}_{t-1} \cdot S_t$
    If $\mathcal{K}_t \geq 1/\alpha$ then stop and **reject** the null
**end for**
**if** the null has not been rejected **then**
    Draw $U \sim \mathrm{Unif}(0,1)$, **reject** if $\mathcal{K}_\tau \geq U/\alpha$
**end if**

Figure 1: **Left:** Description of main algorithm. **Right:** The wealth process $(\mathcal{K}_t)$ under different conditions on the means. Observations follow a Bernoulli distribution for a given mean. As $\Delta = |\mu_0 - \mu_1|$ increases, the wealth grows more quickly. If a test rejects, the procedure ends, hence the plateauing of the blue and purple lines. For $\Delta = 0$, the wealth fluctuates around 1 and the test never rejects. Shaded regions indicated the standard deviation after 100 trials.

increases, the wealth grows more quickly which leads to faster rejection of the null. To define ONS, let $g_t = \widehat{Y}_t^0 - \widehat{Y}_t^1$ and initialize $\lambda_1 = 0$. For all $t \geq 1$, recursively define

$$\lambda_t = \left( \left( \frac{2}{2 - \ln(3)} \frac{z_{t-1}}{1 + \sum_{i=1}^{t-1} z_i^2} - \lambda_{t-1} \right) \wedge 1/2 \right) \vee -1/2, \quad \text{where } z_i = \frac{g_i}{1 - \lambda_i g_i}. \quad (3)$$

The machinery just introduced is sufficient to define a level-$\alpha$ sequential test. However, we add a final ingredient to increase the power of this procedure. Motivated by the *randomized* Ville's inequality mentioned in Section 2, we introduce one final step: If we stop the procedure at some stopping time $\tau$ but have not yet rejected the null (because, for instance, our budget ran out), we can check if $\mathcal{K}_\tau \geq U/\alpha$, where $U$ is uniform on [0,1] and *drawn independently from everything observed so far*. We emphasize that the final step can only be performed once. Thus, if it is possible that more data will be collected in the future, we advise waiting. We summarize the process in Algorithm 1.

The following proposition gives a bound on the expected stopping time of this sequential test under the alternative. We note that Shekhar and Ramdas [28] also provide a bound on the expected stopping time of a multivariate betting-style sequential test. However, due to the generality of their setting, their result is defined in terms of quantities which are difficult to analyze. We thus provide a more direct analysis specific to the difference of univariate means. The proof may be found in Appendix C.1.

**Proposition 1.** *Algorithm 1 with input $\alpha \in (0,1)$ and betting strategy (2) is a level-$\alpha$ sequential test with asymptotic power one. Moreover, letting $\Delta = |\mu_0 - \mu_1|$, under the alternative the expected stopping time $\tau$ obeys*

$$\mathbb{E}[\tau] \lesssim \frac{1}{\Delta^2} \log\left(\frac{1}{\Delta^2 \alpha}\right). \quad (4)$$

It is possible to demonstrate a lower bound of $\mathbb{E}[\tau] \gtrsim \sigma^2 \log(1/\alpha)/\Delta^2$ where $\sigma^2 = \mathbb{E}[(\widehat{Y}^0 - \widehat{Y}^1)^2]$ [28, Prop. 2]. Since $\sigma^2 \gtrsim 1$ in the worst case, our result is optimal up to a factor of $\log(1/\Delta^2)$.

### 3.2 Time-varying data collection policies

Here we extend the setting to allow for time-dependent data collection policies. This is motivated by the fact that organizations are often collecting data for various purposes and must therefore evaluate fairness on data which is not necessarily representative of the population at large. We also allow the data collection policies to differ by group. Mathematically, for $b \in \{0, 1\}$ let $\pi_t^b(\cdot) = \pi_t(\cdot | \xi_b)$ denote the randomized policies which determine the probability with which a set of covariates are

selected from the population. Let $\rho^b$ be the density of $X|\xi_b$. Following standard propensity weighting techniques [46, 47], introduce the weighted estimates

$$\omega_t^b(x) := \frac{\rho^b(x)}{\pi_t^b(x)}. \tag{5}$$

We may write simply $\omega_t^b$ when $x$ is understood from context. The policy $\pi_t^b$ (and hence the weight $\omega_t^b$) need not be deterministic, only $\mathcal{F}_{t-1}$ measurable (so it can be considered a deterministic quantity at time $t$). This is motivated by various practical applications of sequential decision-making, in which the policy used to collect data often changes over time in response to either learning or policy [48, 49]. In many applications, $\pi_t^b$ is a function of a select few covariates only—for instance, income, education level, or career. In some settings it is reasonable to assume knowledge of $\rho^b$ (by means of a census, for instance). However, this is not always the case. Section 3.3 will therefore discuss strategies which do not require precise knowledge of the density. We will assume that for all $x, t$ and $b$, $\omega_t^b(x) < \infty$. In this case, (5) enables an unbiased estimate of $\mu_b = \mathbb{E}_\rho[\varphi(X)|\xi_b]$ when $X_t$ is sampled according to $\pi_t^b$:

$$\mathbb{E}_{X \sim \pi_t^b}[\widehat{Y}_t^b \omega_t^b(X)|\mathcal{F}_{t-1}, \xi_b] = \int_{\mathcal{X}} \varphi(x)\omega_t^b(x)\pi_t^b(x)\mathrm{d}x = \int_{\mathcal{X}} \varphi(x)\rho^b(x)\mathrm{d}x = \mu_b. \tag{6}$$

Our payoff function is similar to the previous section, but reweights the samples by $\omega_t^b(=\omega_t^b(X_t^b))$, and then adds a corrective factor to ensure that the payoff is nonnegative:

$$S_t(\widehat{Y}_t^0, \widehat{Y}_t^1) = 1 + \lambda_t L_t(\widehat{Y}_t^0 \omega_t^0 - \widehat{Y}_t^1 \omega_t^1), \quad \text{where} \quad L_t := \min_{b \in \{0,1\}} \operatorname*{ess\,inf}_{x \in \mathcal{X}} \frac{1}{2\omega_t^b(x)}, \tag{7}$$

and $\lambda_t$ is once again selected via ONS. Multiplication by $L_t$ is required to ensure that the values $L_t(\widehat{Y}_t^0 \omega_t^0 - \widehat{Y}_t^1 \omega_t^1)$ lie in $[-1, 1]$ in order to be compatible with ONS. Here "ess inf" is the essential infimum, which is the infimum over events with nonzero measure. For most practical applications one can simply replace this with a minimum. For such a strategy, we obtain the following guarantee.

**Proposition 2.** *Algorithm 1 with input $\alpha \in (0, 1)$ and betting strategy (7) is a level-$\alpha$ sequential test with asymptotic power one. Moreover, suppose that $0 < L_{\inf} := \inf_{t \geq 1} L_t$ and let $\kappa = \Delta L_{\inf}$ and $\Delta = |\mu_0 - \mu_1|$. Then, treating log-log factors as constant, under the alternative the expected stopping time $\tau$ obeys*

$$\mathbb{E}[\tau] \lesssim \frac{1}{\kappa^2} \log\left(\frac{1}{\kappa^2 \alpha}\right). \tag{8}$$

Thus we see that we pay the price of allowing randomized data collection policies by a factor of roughly $1/L_{\inf}$. Note that the payoff function (7) does not require knowledge of $L_{\inf}$. Indeed, this strategy remains valid even if $L_{\inf} = 0$ as long as $L_t$ is nonzero for each $t$. It is only the analysis that requires $L_{\inf}$ to be finite and known. The proof of Proposition 2 is provided in Appendix C.2.

### 3.3 Time-varying policies for unknown densities

A reasonable objection to the discussion in the previous section is that the densities $\rho^b$ may not always be known. In this case we cannot compute the propensity weights in (5). Here we provide an alternative payoff function which uses an estimate of the density, $\widehat{\rho}^b$. We assume we know upper and lower bounds on the multiplicative error of our estimate: $\delta^{\max} \geq \max_b \sup_x \{\widehat{\rho}^b(x)/\rho^b(x)\}$, and $\delta^{\min} \leq \min_b \inf_x \{\widehat{\rho}^b(x)/\rho^b(x)\}$. We assume that $\delta^{\min} > 0$.

Let $\widehat{\omega}_t^b$ be the propensity weights at time $t$ using the estimated density, i.e., $\widehat{\omega}_t^b(x) = \widehat{\rho}^b(x)/\pi_t^b(x)$. Notice that $\mathbb{E}_{X \sim \pi_t^b}[\widehat{\omega}_t^b(X)\varphi(X)|\mathcal{F}_{t-1}] = \int_{\mathcal{X}} \varphi(x)\widehat{\rho}^b(x)\mathrm{d}x = \int_{\mathcal{X}} \varphi(x)(\widehat{\rho}^b(x)/\rho^b(x))\rho^b(x)\mathrm{d}x \leq \delta^{\max}\mu_b$. Similarly, $\mathbb{E}_{X \sim \pi_t^b}[\widehat{\omega}^b(X)\varphi(X)|\mathcal{F}_{t-1}, \xi_b] \geq \delta^{\min}\mu_b$. Recall that $\widehat{Y}_t^b = \varphi(X_t^b)$. Consider the payoff function

$$S_t(\widehat{Y}_t^0, \widehat{Y}_t^1) = 1 + \lambda_t B_t\left(\frac{\widehat{\omega}_t^0 \widehat{Y}_t^0}{\delta^{\max}} - \frac{\widehat{\omega}_t^1 \widehat{Y}_t^1}{\delta^{\min}}\right), \tag{9}$$

where $B_t$ is $\mathcal{F}_{t-1}$ measurable. Then

$$\mathbb{E}[S_t|\mathcal{F}_{t-1}] = 1 + \lambda_t B_t\left(\frac{\mathbb{E}_{\pi_t^0}[\widehat{\omega}_t^0 \widehat{Y}_t^0|\mathcal{F}_{t-1}]}{\delta^{\max}} - \frac{\mathbb{E}_{\pi_t^1}[\widehat{\omega}_t^1 \widehat{Y}_t^1|\mathcal{F}_{t-1}]}{\delta^{\min}}\right) \leq 1 + \lambda_t B_t(\mu_0 - \mu_1).$$

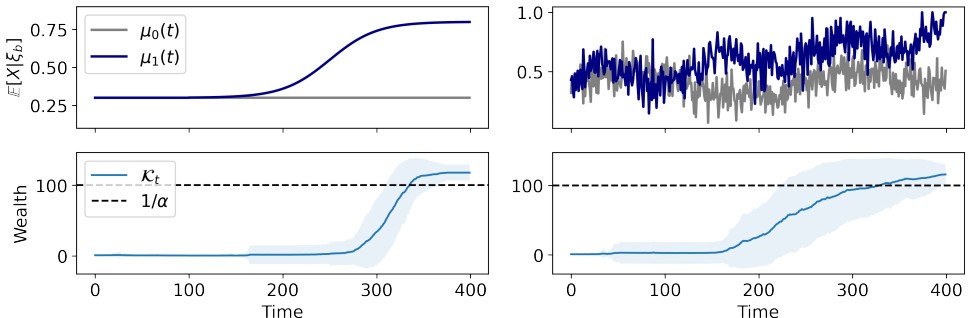

Figure 2: Two illustrations of our sequential test adapting to distribution shift. In both settings, the observations at time $t$ are Bernoulli with bias determined by the respective mean at that time. Shaded areas in the bottom plots represent the standard deviation across 100 trials. **Left:** For the first 100 time steps, we have $\mu_0(t) = \mu_1(t) = 0.3$. At time 100, $\mu_1(t)$ begins to smoothly slope upward. **Right:** Here we assume that both means are non-stationary and non-smooth. Both are sinusoidal with Gaussian noise but $\mu_1(t)$ drifts slowly upwards over time.

Under the null, $\mu_0 - \mu_1 = 0$, implying that $\mathbb{E}[S_t | \mathcal{F}_{t-1}] \leq 1$. We select $B_t$ so that $S_t$ is nonnegative (thus ensuring that the capital process is a nonnegative supermartingale) and is compatible with ONS. The following condition on $B_t$ suffices: $B_t \leq \delta^{\min} \min_b \inf_x (2\widehat{\omega}_t^b(x))^{-1}$. Note that if we know the true densities then we can take $\delta^{\max} = \delta^{\min} = 1$ and we recover (7).

### 3.4 Handling distribution shift

Until this point we've assumed that the means $\mu_b$ remain constant over time. A deployed model, however, is susceptible to distribution shift due to changes in the underlying population, changes in the model (retraining, active learning, etc), or both. Here we demonstrate that our framework handles both kinds of drift. In fact, we need not modify our strategy (we still deploy Algorithm 1), but we must reformulate our null and alternative hypotheses and our analysis will change. Before introducing the mathematical formalities, we draw the reader's attention to Figure 2 which illustrates two cases of distribution shift and the response of our sequential test to each. The left panel is a case of "smooth" drift in which case $\mu_0(t) = \mu_1(t)$ are equal for some number of timesteps after which $\mu_1(t)$ begins to drift upward. The right panel exemplifies a situation in which both means change each timestep (the weekly prevalence of disease in a population, for instance), but $\mu_1(t)$ has a marked drift upward over time. In both cases, the wealth process is sensitive to the drift and grows over time.

To handle changes in the underlying population we let $\rho_t$ denote the population distribution over $\mathcal{X}$ at time $t$. Likewise, to handle changes to the model, we let $\varphi_t : \mathcal{X} \to [0, 1]$ be the model at time $t$. In order to define our hypotheses, let $\Delta_t := |\mu_0(t) - \mu_1(t)|$ where $\mu_b(t) = \mathbb{E}_{X \sim \rho_t}[\varphi_t(X) | \xi_b, \mathcal{F}_{t-1}]$, $b \in \{0, 1\}$. Then, we write the null and alternative hypothesis as

$$H_0 : \Delta_t = 0 \ \forall t \geq 1, \quad H_1 : \exists T \in \mathbb{N} \ \text{s.t.} \ \Delta_t > 0 \ \forall t \geq T. \tag{10}$$

Of course, $H_0$ and $H_1$ do not cover the space of possibilities. In particular, neither contains the event that $\mu_1(t) \neq \mu_0(t)$ for some finite interval $t \in [a, b]$ and are otherwise equal. However, we believe (10) strikes a desirable balance between analytical tractability and practical utility; defining $H_1$ such that we will reject the null for each finite window $[a, b]$ places too extreme a burden on the growth of the wealth process. Moreover, as illustrated by Figure 2, if $\Delta_t > 0$ for a large enough window, then our sequential tests will reject the null.

If we are working with randomized data collection policies per Section 3.2, then one should change the definition of the weights in Equation (5) to incorporate the time-varying distribution $\rho_t$. That is, $\omega_t^b(x) = \rho_t^b(x)/\pi_t^b(x)$. Otherwise, all procedures detailed in the previous sections remain the same, and we can provide the following guarantee on their performance.

**Proposition 3.** *Algorithm 1 with input $\alpha \in (0, 1)$ and betting strategies (2) and (7) is a level-$\alpha$ sequential test for problem (10). It has power one under the alternative if $\Delta_{\inf} := \inf_{t \geq n} \Delta_t > 0$ where $n$ is some time at which drift begins. Moreover, under the alternative the expected stopping*

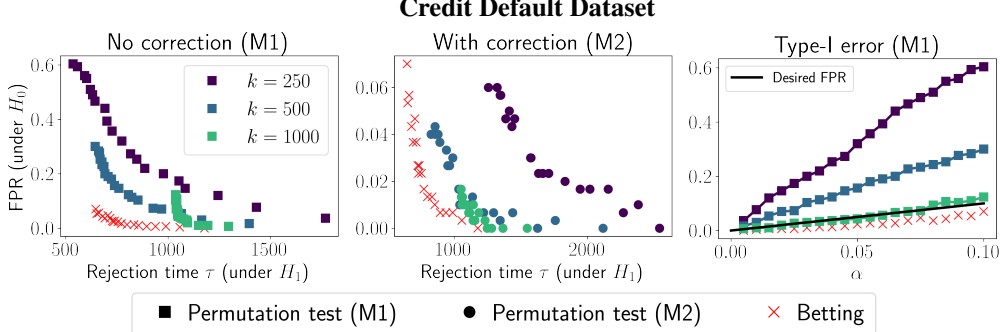

Figure 3: Comparisons of false positives rates (FPRs) and stopping times on credit loan data and US census data. The left two columns plot $\tau$ under $H_1$ versus the FPR as $\alpha$ is varied from 0.1 to 0.01. The FPR is grossly inflated when under method M1, as illustrated the first and third columns. Betting is a Pareto improvement over the permutation tests.

*time $\tau$ obeys*

$$\mathbb{E}[\tau] \lesssim n + \frac{1}{\Delta_{\inf}^2} \log\left(\frac{1}{\Delta_{\inf}^2 \alpha}\right). \tag{11}$$

*Furthermore, if the data are gathered by randomized policies $\pi_t^b$ and $L_{\inf} > 0$ for $L_{\inf}$ as in Proposition 2, then the expected stopping time follows by replacing $\Delta_{\inf}$ in (14) with $\Delta_{\inf} L_{\inf}$.*

Let us make two remarks about this result. First, the condition $\Delta_{\inf} > 0$ ensures that, after some point, the means $\mu_0(t)$ and $\mu_1(t)$ remain separated. If they diverge only to later reconverge, there is no guarantee that the test will reject (though, in practice, it will if they are separated for sufficiently long). Second, the reader may have guessed that the expected stopping time under drift beginning at time $n$ follows from Propositions 1 and 2 after simply adding $n$. However, it is a priori feasible that the reliance of ONS on past predictions would result in slower convergence under distribution shift. Proposition 3 verifies that this is not the case, and that the rate is the same up to constants.

### 3.5 Composite Nulls

In practice, it may be unreasonable to require that the means between two groups are precisely zero. Instead, we may only be interested in detecting differences greater than $\epsilon$ for some $0 < \epsilon \ll 1$. In this case we may write the null and alternative as

$$H_0 : |\mu_0 - \mu_1| \le \epsilon \text{ vs } H_1 : |\mu_0 - \mu_1| > \epsilon. \tag{12}$$

To formulate our test, we introduce two pairs of auxiliary hypotheses: $H_0' : \mu_0 - \mu_1 \le \epsilon$ vs $H_1' : \mu_0 - \mu_1 > \epsilon$ and $H_0'' : \mu_1 - \mu_0 \le \epsilon$ vs $H_1'' : \mu_1 - \mu_0 > \epsilon$. Observe that if either $H_0'$ or $H_0''$ is false, then the alternative $H_1$ is true. Our approach will therefore entail testing both $H_0'$ vs $H_1'$ and $H_0''$ vs $H_1''$. Let $\phi_t^Q$ denote the sequential test for the former, and $\phi_t^R$ for the latter. The test given by $\phi_t = \max\{\phi_t^Q, \phi_t^R\}$ (i.e., rejecting $H_0$ if *either* $H_0'$ or $H_0''$ is rejected) is then a test for (12). Game-theoretically, this can be interpreted as splitting our initial capital in half and playing two games simultaneously. To test $H_0'$ and $H_0''$ consider the two payoff functions

$$Q_t(\widehat{Y}_t^0, \widehat{Y}_t^1) = 1 + \lambda_t(\widehat{Y}_t^0 - \widehat{Y}_t^1 - \epsilon), \text{ and } R_t(\widehat{Y}_t^0, \widehat{Y}_t^1) = 1 + \lambda_t(\widehat{Y}_t^1 - \widehat{Y}_t^0 - \epsilon). \tag{13}$$

If $\lambda_t \in [\frac{-1}{1-\epsilon}, \frac{1}{1+\epsilon}]$ then both $Q_t$ and $R_t$ are nonnegative. We thus select $\lambda_t$ via ONS as usual. The wealth processes $(\mathcal{K}_t^Q)$ and $(\mathcal{K}_t^R)$ defined by $Q_t$ and $R_t$ (i.e., $\mathcal{K}_t^Q = \prod_{i \le t} Q_i(\widehat{Y}_i^0, \widehat{Y}_i^1)$ and $\mathcal{K}_t^R = \prod_{i \le t} R_i(\widehat{Y}_i^0, \widehat{Y}_i^1)$) are then nonnegative supermartingales under $H_0'$ and $H_0''$, respectively. We reject $H_0$ if either $\mathcal{K}_t^R \ge 2/\alpha$ or $\mathcal{K}_t^Q \ge 2/\alpha$ which results in an overall level-$\alpha$ sequential test.

**Proposition 4.** *Algorithm 1 with input $\alpha/2 \in (0,1)$ with betting strategy $Q_t$ (resp., $R_t$) is a level-$\alpha/2$ sequential test for $H_0'$ vs $H_1'$ (resp., $H_0''$ vs $H_1''$). Rejecting $H_0$ if either $H_0'$ or $H_0''$ is rejected results in a level-$\alpha$ sequential test with asymptotic power one for problem (12). Let $\Delta = |\mu_0 - \mu_1|$. Then, under the alternative the expected stopping time $\tau$ obeys*

$$\mathbb{E}[\tau] \lesssim \frac{1}{(\Delta - \epsilon)^2} \log\left(\frac{1}{\alpha(\Delta - \epsilon)^2}\right). \tag{14}$$

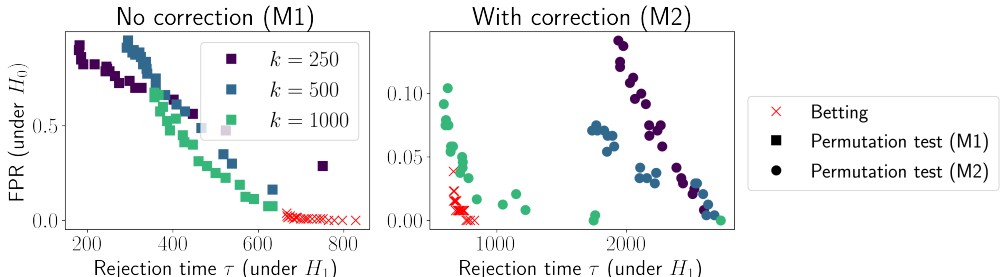

Figure 4: Response of the tests to distribution shift on the census data. We use a fair model for 400 timesteps, after which we switch to an unfair model with $\Delta \approx 1.0$. The leftmost plot uses permutation tests under M1, resulting in inflated type-I error. Values are plotted as $\alpha$ ranges from 0.01 to 0.1.

We note that under the alternative, $\epsilon < \Delta$, so $0 < \Delta - \epsilon < \Delta$. Proposition 4 states that the expected stopping time depends on the interplay of $\Delta$ and $\epsilon$, and will increase as they approach each other. This is intuitive. For a fixed $\Delta$ one would expect that the problem becomes harder as $\epsilon \uparrow \Delta$.

## 4  Experiments

Here we provide several sets of experiments to demonstrate the benefits of our sequential test compared to fixed-time tests.[2] First we need to consider how fixed-time tests might be applied in practice to deal with sequential settings. We consider two such methods.

**M1.** For some prespecified $k \in \mathbb{N}$, wait until we have collected $k$ audits, then perform the test. If the test does not reject, collect another $k$ audits and repeat. We emphasize that if one does not adjust the significance level over time, then *this is not a valid level-$\alpha$ test*. However, it may be used unwittingly in practice, so we study it as one baseline.

**M2.** We batch and test in the same way as above, but we apply a Bonferroni-like correction in order to ensure it is level $\alpha$. More precisely, for the $j$th-batch, $j \geq 1$, we set the significance level to $\alpha/2^j$. The union bound then ensures that the FPR is at most $\alpha$ over all batches.

We run experiments on three real world datasets: a credit default dataset [50], US census data [51], and health insurance data [52]. All can be made to violate equality of opportunity if trained with naive models; we provide the details in Appendix D. Suffice it to say here that the absolute difference in means, $\Delta$, is $\approx 0.03$ for the credit default dataset, $\approx 1.1$ for the census data, and $\approx 0.06$ for the insurance data. We employ various distinct models to test our methods, from forest-based models to logistic regression. We use a permutation test as our baseline fixed-time test because of its ubiquity in practice, exact type-I error control under exchangeability, and minimax optimality in various scenarios [53].

Figure 3 compares the results of Algorithm 1 (betting strategy (2)) with the permutation test baselines M1 and M2. The left two columns plot the empirical false positive rate (FPR) against the stopping time under the alternative. Values concentrated in the lower left corner are therefore preferable. We plot values for each test as $\alpha$ is varied from 0.01 to 0.1. The betting test typically Pareto-dominates the (stopping time, FPR) values achievable by the baselines. In a small number of cases, M1 (baseline without Bonferroni correction) achieves a faster stopping time; however, this is only possible with a very high FPR (> 0.5). This inflated FPR is verified by the final column, which shows that M1 does not respect the desired type-I error rate. This encourages the use of the Bonferroni-style correction of M2. However, doing so results in an overly conservative test which is slower to reject than betting, as seen in the center column of Figure 3. Betting, meanwhile, respects the desired FPR over all $\alpha$, and almost always has the fastest stopping time at any given FPR. We note that while it may appear that the permutation tests improve as $k$ gets larger, this trend does not hold for large $k$. Indeed, $k$ is

---

[2]We refrain from comparing our betting-style sequential tests to other *sequential* tests; we refer the reader to [28] for such comparisons. Instead, our goal in this section is to persuade the fairness community that sequential tests are superior tools to fixed-time tests for auditing deployed systems.

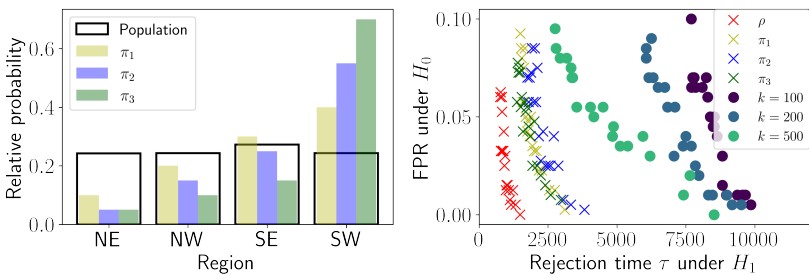

Figure 5: Illustration of our betting method when using various data collection strategies, $\pi_1$, $\pi_2$, and $\pi_3$, which were based on each individual's region (NE, NW, SE, SW). Operationally, $\pi_i$ samples an individual by first sampling a region with the given probability, and then sampling an individual uniformly at random from that region. We compare results against our method with data sampled uniformly from the population (red crosses), and to permutation tests (M2), also with uniformly sampled data. Even with randomized policies, we continue to outperform permutation tests.

also the minimum rejection time for the permutation test. Increasing $k$ to 1,500 on the credit default dataset, for instance, would result in vertical line at $\tau = 1,500$.

The robustness of betting to distribution shift is illustrated by Figure 4. Here we use a fair model for the first 400 timesteps, after which we switch to a random forest classifier which is unfair ($\Delta \approx 1.1$). The rejection time for all tests suffers accordingly, but betting remains a Pareto improvement over the permutation tests using M2. As before, using M1 results in an enormous FPR. Interestingly, we see that M2 also results in FPR higher than 0.1 (the maximum value of $\alpha$) a nontrivial percentage of the time. Indeed, under distribution shift, the the permutation test using M2 is no longer level-$\alpha$ since the data are not exchangeable. Betting, on the other hand, maintains its coverage guarantee.

Finally, Figure 5 illustrates the performance of various algorithms when audits are not conducted on predictions received uniformly at random from the population (discussed in Section 3.2). Instead, predictions were received based on an individual's region. We test three distinct strategies, $\pi_1$, $\pi_2$ and $\pi_3$, each of which deviates from the population distribution to varying degrees (LHS of Figure 5). As expected, the rejection time of our strategy suffers compared to when the predictions are received uniformly from the population (denoted by the red crosses). The desired level-$\alpha$ guarantee is still met, however, and betting continues to outperform permutation tests in all cases.

## 5 Summary

We have argued that practitioners in the field of algorithmic fairness should consider adopting sequential hypothesis tests in lieu of fixed-time tests. The former enjoy two desirable properties absent in the latter: (i) the ability to continually monitor incoming data, and (ii) the ability to reject at data-dependent stopping times. Both (i) and (ii) are useful for high-stakes and/or time-sensitive applications in which fixing the budget beforehand and waiting for more data to arrive may be unsatisfactory, such as auditing the fairness of healthcare models [54] or resource allocation strategies during a public health crisis [55, 56]. We provided a sequential test with asymptotic power one for group fairness (Algorithm 1) inspired by the paradigm of game-theoretic probability and statistics [25, 21, 28]. It is fast, easy to implement, and can handle time-varying data collection policies, distribution shift, and composite nulls, all of which are common in practice. We also provided bounds on the expected stopping time of the test. We hope that the simplicity of our methods combined with their strong theoretical guarantees prove useful for practitioners in the field.

**Limitations and societal impact.** Auditing is a complex socio-technical problem which involves a range of questions: choosing appropriate audit targets, gaining access to data, ensuring the credibility and independence of external auditors, creating accountability based on audit results, and more [57]. Indeed, poorly designed audits may certify that a system is "fair" while masking harder to verify issues such as data provenance. Our work addresses only one portion of the overall auditing task: providing statistically sound methods for the sequential testing of a specific criteria. This is complementary to, and does not substitute for, careful overall design of an auditing framework.

**Acknowledgements.** We thank Jing Yu Koh and Shubhanshu Shekhar for helpful conversations. We also thank the anonymous referees for helpful feedback which improved the paper. BC and AR acknowledge support from NSF grants IIS-2229881 and DMS-2310718. BC was supported in part by the NSERC PGS D program, grant no. 567944. BW and SCG were supported in part by the AI2050 program at Schmidt Futures, grant no. G-22-64474.

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

# A  Testing in batches

Of course, it is the rare application in which we can expect to receive an audit from each group at each timestep. We can modify the strategy in Section 3 to handle different arrival times by simply waiting to bet until we have new audits from each group. If multiple audits from the other group have accumulated during that period, we bet with their average. More formally, define

$$N_t^b = T_b[t] \cap \{n : n \geq \max(T_{1-b}[t])\}, \text{ and } G_t^b = \frac{1}{|N_t^b|} \sum_{j \in N_t^b} \widehat{Y}_j^b, \tag{15}$$

where we recall that $T_b[t] = T_b \cap [t]$ is the set of times at which we receive audits from group $b$ up until time $t$. In words, $N_t^b$ is simply all timesteps for which we receive an audit from group $b$ without receiving an audit from the other group. From here, we can define a payoff function as

$$S_t = \begin{cases} 1 + \lambda_t(G_t^0 - G_t^1), & \text{if } N_t^0 \neq \emptyset \text{ and } N_t^1 \neq \emptyset, \\ 1, & \text{otherwise.} \end{cases} \tag{16}$$

If $S_t = 1$ then $\mathcal{K}_t = \mathcal{K}_{t-1}$, so this can be interpreted as abstaining from betting at time $t$. This occurs when we have no new audits from one group ($N_t^b = \emptyset$ for some $b$). Note that if $N_t^0$ and $N_t^1$ are both non-empty, then one of them contains only $t$. Thus, in this case, $G_t^0 - G_t^1 \in \{\widehat{Y}_t^0 - G_t^1, G_t^0 - \widehat{Y}_t^1\}$. As in Section 3 we choose $\lambda_t$ using ONS, but we use $g_t = G_t^0 - G_t^1$ in its definition. The expected stopping time of the test defined by (16) follows as a corollary from Proposition 1 after correcting for the number of timesteps during which non-trivial bets were placed. More specifically, the stopping-time in Proposition 1 is with reference to the number of times we actually bet, i.e., with respect to the times $\{t : N_t^0 \neq \emptyset, N_t^1 \neq \emptyset\}$. The same remark holds for other propositions throughout this paper and, as such, we will state these propositions under the assumption that $T_0 = T_1 = \mathbb{N}$.

Finally, let us emphasize that the above discussion is still assuming an online data gathering procedure. If we are in a nonsequential setting (i.e., all the data has already been gathered), then we may simulate a sequential setting by simply revealing the outcomes one at a time. Thus, all of our procedures hold for fixed-time hypothesis testing as well.

# B  Auditing multiple groups

Here we consider the case when there are more than two groups. Suppose we have $J + 1$ groups $\{0, 1, \ldots, J\}$. In accordance with Definition 1, the null and alternative become

$$H_0 : \mathbb{E}_\rho[\varphi(X)|\xi_i] = \mathbb{E}_\rho[\varphi(X)|\xi_j], \quad \forall i, j \in \{0, \ldots, J\}, \tag{17}$$
$$H_1 : \exists i, j \in \{0, \ldots, J\} \text{ such that } \mathbb{E}_\rho[\varphi(X)|\xi_i] \neq \mathbb{E}_\rho[\varphi(X)|\xi_j]. \tag{18}$$

As before, let $\mu_i = \mathbb{E}_\rho[\varphi(X)|\xi_i]$, $i \in \{0, \ldots, J\}$. One could derive a sequential test by applying Algorithm 1 to each pair of means $\mu_i$ and $\mu_j$. Game-theoretically, this can be interpreted as splitting your initial wealth among multiple games and playing each simultaneously. If you grow rich enough in any one game, you reject the null. Of course, one needs to adjust the significance level to account for the number of games being played, thus reducing the (nonasymptotic) power of the test.

Of course, it is not necessary to test each mean against all others. We need only test whether $\mu_b = \mu_{b+1}$ for all $b \in \{0, \ldots, J\}$. That is, we can play $J$ games instead of $\Omega(J^2)$ games. In order to ensure this constitutes a level-$\alpha$ test, we reject when the wealth process of any game is at least $J/\alpha$. For any single game, this occurs with probability $\alpha/J$ under the null. Therefore, the union bound then ensures that the type-I error of this procedure is bounded by $\alpha$. Moreover, the asymptotic power remains one since, if $\mu_i \neq \mu_j$ for some $i, j$ then $\mu_b \neq \mu_{b+1}$ for some $b$. The guarantees we've provided on Algorithm 1 ensure that the wealth process for this particular game will eventually grow larger than $J/\alpha$ and our test will reject. We summarize this discussion with the following proposition, which is the equivalent of Proposition 1 for auditing multiple groups.

**Proposition 5.** *Let $\alpha \in (0, 1)$. Consider running Algorithm 1 on groups $b, b + 1$, for $b \in \{0, 1, \ldots, J - 1\}$ in parallel with input parameter $\alpha/J$. This constitutes a level-$\alpha$ sequential test for (17) with asymptotic power one against (18). If we receive an audit from each group at each timestep, then the expected stopping time $\tau$ of this procedure obeys*

$$\mathbb{E}[\tau] \lesssim \min_{b \in \{0, \ldots, J-1\}} \frac{1}{|\mu_b - \mu_{b+1}|^2} \log\left(\frac{J}{|\mu_b - \mu_{b+1}|^2 \alpha}\right). \tag{19}$$

The expected stopping time follows from Proposition 1 after correcting for the significance level and the difference between the means. We take the minimum over all $b$ because the procedure rejects as soon as *any* of the wealth processes grow too large. Equivalent versions of Propositions 2 and 3 for multiple groups can be obtained similarly.

Let us end by remarking that similar ideas may be applied to test other notions of group fairness which posit the equivalence of multiple means. This is the case of equalized odds, for instance. As above we simply imagine playing multiple games simultaneously, each testing for the equivalence of one pair of means.

## C  Omitted proofs

### C.1  Proof of Proposition 1

We break the proof into three components.

**Level-$\alpha$ sequential test.** Combining the discussion at the beginning of Section 3 with Ville's inequality demonstrates why our procedure constitutes a level-$\alpha$ sequential test. However, let us prove it formally here for completeness. Let $P \in H_0$ and note that $\mathbb{E}_P[\widehat{Y}_t^0 - \widehat{Y}_t^1] = \mathbb{E}_P[\varphi(X_t^0) - \varphi(X_t^1)] = \mu_0 - \mu_1 = 0$. Therefore, using the fact that $\lambda_t$ is predictable (i.e., $\mathcal{F}_{t-1}$-measurable)

$$\mathbb{E}_P[\mathcal{K}_t | \mathcal{F}_{t-1}] = \mathbb{E}_P\left[\prod_{j=1}^t (1 + \lambda_j(\widehat{Y}_j^0 - \widehat{Y}_j^1)) \Big| \mathcal{F}_{t-1}\right] = \mathcal{K}_{t-1}(1 + \lambda_t \mathbb{E}_P[\widehat{Y}_t^0 - \widehat{Y}_t^1]) = \mathcal{K}_{t-1},$$

so $(\mathcal{K}_t)_{t \geq 1}$ is a $P$-martingale, with initial value 1. Moreover, it is nonnegative since $|\lambda_t| \leq 1/2$ for all $t$ by definition of ONS. Thus, Ville's inequality implies $P(\exists t \geq 1 : \mathcal{K}_t \geq 1/\alpha) \leq \alpha$, meaning that rejecting at $1/\alpha$ yields a level-$\alpha$ sequential test. Finally, as discussed in the main paper, the last lines of Algorithm 1 are justified by the randomized Ville's inequality of Ramdas and Manole [43], which states that, for all stopping times $n$,

$$P(\exists t \leq n : \mathcal{K}_t \geq 1/\alpha \ \text{ or } \ \mathcal{K}_n > U/\alpha) \leq \alpha,$$

where $U \sim \text{Unif}(0,1)$ is independent of everything else.

**Asymptotic power.** Next, let us demonstrate that Algorithm 1 has asymptotic power one. That is, for $P \in H_1$, $P(\tau < \infty) = 1$. It suffices to show that $P(\tau = \infty) = 0$. To see this, define

$$g_t := \widehat{Y}_t^0 - \widehat{Y}_t^1, \quad S_t := \sum_{i=1}^t g_i, \quad V_t := \sum_{i=1}^t g_i^2.$$

We have the following guarantee on the wealth process, which can be translated from results concerning ONS from Cutkosky and Orabona [44, Theorem 1]:

$$\mathcal{K}_t \geq \frac{1}{V_t} \exp\left\{\frac{S_t^2}{4(V_t + |S_t|)}\right\} \geq \frac{1}{t} \exp\left\{\frac{S_t^2}{8t}\right\}, \quad \forall t \geq 1. \tag{20}$$

Since $\{\tau = \infty\} \subset \{\tau \geq t\}$ for all $t \geq 1$, we have $P(\tau = \infty) \leq \liminf_{t \to \infty} P(\tau > t) \leq \liminf_{t \to \infty} P(\mathcal{K}_t < 1/\alpha)$, where the final inequality is by definition of the algorithm. Using the second inequality of (20),

$$P(\mathcal{K}_t < 1/\alpha) \leq P\left(\exp\left\{\frac{S_t^2}{8t}\right\} < t/\alpha\right) \leq P\left(-\sqrt{\frac{8\log(t/\alpha)}{t}} < \frac{S_t}{t} < \sqrt{\frac{8\log(t/\alpha)}{t}}\right).$$

By the SLLN, $S_t/t$ converges to $\mu_0 - \mu_1 \neq 0$ almost surely. On the other hand, $8\log(t/\alpha)/t \to 0$. Thus, if we let $A_t$ be the event that $\exp(S_t^2/8t) < t/\alpha$, we see that $\mathbf{1}(A_t) \to 0$ almost surely. Hence, by the dominated convergence theorem,

$$P(\tau = \infty) \leq \liminf_{t \to \infty} P(A_t) = \liminf_{t \to \infty} \int \mathbf{1}(A_t) dP = \int \liminf_{t \to \infty} \mathbf{1}(A_t) dP = 0.$$

This completes the argument.

**Expected stopping time.** Last, let us show the desired bound on the expected stopping time. Fix $P \in H_1$. Let $\tau$ be the stopping time of the test. Since it is nonnegative, we have

$$\mathbb{E}[\tau] = \sum_{t=1}^{\infty} P(\tau > t) = \sum_{t=1}^{\infty} P(\log \mathcal{K}_t < \log(1/\alpha)) = \sum_{t=1}^{\infty} P(E_t),$$

for $E_t = \{\log \mathcal{K}_t < \log(1/\alpha)\}$. Note that the second equality is by definition of the algorithm. Employing the first inequality of (20) yields

$$\begin{aligned} E_t &\subset \{S_t^2 < 4(V_t + |S_t|)(\log(1/\alpha) - \log(1/V_t))\} \\ &\subset \left\{ S_t^2 < 4\left( V_t + \sum_{i \le t} |g_i| \right)(\log(1/\alpha) - \log(1/V_t)) \right\}. \end{aligned}$$

To analyze the probability of this event, we first develop upper bounds on $W_t := \sum_{i \le t} |g_i|$ and $V_t$. We begin with $W_t$. Since $W_t$ is the sum of independent random variables in $[0, 1]$, we apply the multiplicative Chernoff bound (e.g., [58]) to obtain

$$P(W_t > (1 + \delta)\mathbb{E}[W_t]) \le \exp(-\delta^2 \mathbb{E}[W_t]/3).$$

Setting the right hand side equal to $1/t^2$ and solving for $\delta$ gives $\delta = \sqrt{6 \log t / \mathbb{E} W_t}$. Thus, with probability $1 - 1/t^2$, we have

$$W_t \le \mathbb{E} W_t + \sqrt{6 \mathbb{E}[W_t] \log t} = t + \sqrt{6t \log t} \le 2t \quad \forall t \ge 17, \tag{21}$$

where we've used that $\mathbb{E}[W_t] = \sum_{i \le t} \mathbb{E}[|g_i|] \le t$ since $|g_i| \le 1$. Following a nearly identical process for $V_t$, we have that with probability $1 - 1/t^2$,

$$V_t \le \mathbb{E}[V_t] + \sqrt{6 \mathbb{E}[V_t] \log t} \le t + \sqrt{6t \log t} \le 2t, \quad \forall t \ge 17, \tag{22}$$

where again we use that $|g_i|^2 \le |g_i| \le 1$. Let $H_t = \{V_t \le 2t\} \cap \{W_t \le 2t\}$. Then,

$$E_t \cap H_t \subset \{S_t^2 < 16t(\log(1/\alpha) + \log(2t))\} \subset \{|S_t| < \underbrace{4\sqrt{t \log(2t/\alpha)}}_{:=D}\}.$$

We now argue that $|S_t|$ is unlikely to be so small. Indeed, since $S_t$ is the sum of independent random variables in $[-1, 1]$, applying a Chernoff bound for the third time gives $P(|S_t - \mathbb{E} S_t| \ge u) \le 2 \exp(-u^2/t)$. So, with probability $1 - 1/t^2$, by the reverse triangle inequality,

$$||S_t| - |\mathbb{E} S_t|| \le |S_t - \mathbb{E} S_t| \le \sqrt{t \log 2t^2},$$

implying that,

$$|S_t| \ge |\mathbb{E} S_t| - \sqrt{t \log 2t^2} \ge t\Delta - \sqrt{2t \log 2t}.$$

This final quantity is at least $D$ for all $t \ge \frac{81}{\Delta^2} \log(\frac{162}{\Delta^2 \alpha})$. Now, combining what we've done thus far, by the law of total probability,

$$P(E_t) = P(E_t \cap H_t) + P(E_t | H_t^c) P(H_t^c) \le P(|S_t| < D) + P(H_t^c) \le 3/t^2,$$

and so, for $t$ large enough such that (21), (22), and $S_t > D$ all hold, that is

$$T = \frac{81}{\Delta^2} \log\left( \frac{162}{\Delta^2 \alpha} \right),$$

we have

$$\mathbb{E}[\tau] = \sum_{t \ge 1} P(E_t) \le T + \sum_{t \ge T} \frac{3}{t^2} \le T + \frac{\pi^2}{2}.$$

This completes the proof.

## C.2 Proof of Proposition 2

The proof is similar to that of Proposition 1, so we highlight only the differences.

The wealth process remains a martingale due to the IPW weights (5). Indeed, since $\lambda_t$ and $L_t$ are $\mathcal{F}_{t-1}$ measurable, under the null we have

$$
\mathbb{E}[\mathcal{K}_t|\mathcal{F}_{t-1}] = \mathbb{E}\left[\prod_{j=1}^{t}(1 + \lambda_j L_j(\widehat{Y}_j^0\omega_j^0 - \widehat{Y}_j^1\omega_j^1))\middle|\mathcal{F}_{t-1}\right]
$$
$$
= \mathcal{K}_{t-1}(1 + \lambda_t L_t\mathbb{E}[\widehat{Y}_t^0\omega_t^0 - \widehat{Y}_t^1\omega_t^1]) = \mathcal{K}_{t-1}(1 + \lambda_t L_t(\mu_0 - \mu_1)) = \mathcal{K}_{t-1}.
$$

Moreover, as described in the text, multiplication by $L_t$ ensures that $\mathcal{K}_t$ is nonnegative, since

$$
|L_t||\widehat{Y}_t^0\omega_t^0(X_t^0) - \widehat{Y}_t^1\omega_t^1(X_t^1)| \leq L_t|\widehat{Y}_t^0\omega_t^0(X_t^0)| + L_t|\widehat{Y}_t^1\omega_t^1(X_t^1)|
$$
$$
\leq L_t\omega_t^0(X_t^0) + L_t\omega_t^1(X_t^1) \leq 1,
$$

since

$$
L_t \leq \frac{1}{2\omega_t^b(X_t^b)},
$$

for each $b$ by definition. Therefore, $(\mathcal{K}_t)_{t\geq 1}$ is a nonnegative martingale and, as before, Ville's inequality implies that rejecting at $1/\alpha$ gives a level-$\alpha$ sequential test.

The asymptotic power follows by replacing $g_t$ in Appendix C.1 with

$$
h_t = L_t(\widehat{Y}_t^0\omega_t^0 - \widehat{Y}_t^1\omega_t^1).
$$

Under the alternative, $h_t$ has non-zero expected value, so identical arguments apply.

Regarding, the expected stopping time, we again argue about $h_t$ instead of $g_t$. Since $|h_t| \leq 1$ (see above), the bounds on $V_t$ and $W_t$ remain as they are in the proof of Proposition 1. The bound on $|\mathbb{E}[S_t]|$ is where the proof departs that in Appendix C.1. In this case we have

$$
\mathbb{E}[S_t|\mathcal{F}_{t-1}] = S_{t-1} + L_t\mathbb{E}[\widehat{Y}_t^0\omega_t^0 - \widehat{Y}_t^1\omega_t^1|\mathcal{F}_{t-1}] = S_{t-1} + L_t(\mu_0 - \mu_1).
$$

Therefore,

$$
\mathbb{E}[S_t] = \mathbb{E}[\mathbb{E}[S_t|\mathcal{F}_{t-1}]] = \mathbb{E}[S_{t-1} + L_t(\mu_0 - \mu_1)] = \mathbb{E}[S_{t-1}] + (\mu_0 - \mu_1)\mathbb{E}[L_t].
$$

Induction thus yields

$$
|\mathbb{E}[S_t]| = \left|(\mu_0 - \mu_1)\sum_{i\leq t}\mathbb{E}[L_i]\right| = \Delta\left|\sum_{i\leq t}\mathbb{E}[L_i]\right| \geq \Delta t L_{\inf}.
$$

From here, we may replace $\Delta$ in the proof in Appendix C.1 with $\Delta L_{\inf}$ and the arithmetic remains the same. This yields the desired result.

## C.3 Proof of Proposition 3

Again, the proof mirrors that of Proposition 1 so we highlight only the differences.

First let us ensure that Algorithm 1 yields a level-$\alpha$ sequential test. As before, it suffices to demonstrate that the wealth process is a nonnegative martingale. The time-varying means do not change this fact from before:

$$
\mathbb{E}[\mathcal{K}_t|\mathcal{F}_{t-1}] = \mathbb{E}\left[\prod_{j=1}^{t}(1 + \lambda_j(\widehat{Y}_j^0 - \widehat{Y}_j^1))\middle|\mathcal{F}_{t-1}\right] = \mathcal{K}_{t-1}(1 + \lambda_t\mathbb{E}[\widehat{Y}_t^0 - \widehat{Y}_t^1|\mathcal{F}_{t-1}]) = \mathcal{K}_{t-1},
$$

since, under the null, $\mathbb{E}[Y_t^0|\mathcal{F}_{t-1}] = \mathbb{E}[\varphi(X)|\xi_0, \mathcal{F}_{t-1}] = \mu_0 = \mu_1 = \mathbb{E}[\varphi(X)|\xi_1, \mathcal{F}_{t-1}] = \mathbb{E}[Y_t^1|\mathcal{F}_{t-1}]$. Nonnegativity once again follows from the ONS strategy.

Asymptotic power follows an identical argument as in Appendix C.1, so we focus on expected stopping time. The event $E_t$ remains the same as in Appendix C.1. We again apply a Chernoff bound

to $W_t$ (the values remain independent, even though they are not necessarily identically distributed), and obtain

$$W_t \leq \mathbb{E}W_t + \sqrt{6\mathbb{E}[W_t]\log t} \leq 2t,$$

for $t \geq 17$ with probability $1 - 1/t^2$, since again, $|g_i| \leq 1$ for each $i$. Similarly, $\mathbb{E}V_t \leq 2t$ with probability $1 - 1/t^2$ for $t \geq 17$. Let the shift begin at time $n$, and set $\Delta = \inf_{t \geq n}|\mu_0(t) - \mu_1(t)|$. Then $|\mathbb{E}S_t| \geq (t-n)\Delta$. As above, we want to find $t$ such that

$$|S_t| \geq |\mathbb{E}S_t| - \sqrt{t\log 2t^2} \geq (t-n)\Delta - \sqrt{t\log 2t^2} \geq D.$$

Rearranging and simplifying this final inequality, we see that it suffices for $t$ to satisfy

$$t - n \geq \frac{6}{\Delta}\sqrt{t\log(2t/\alpha)}. \tag{23}$$

We claim this holds for all

$$t \geq n + \max\left\{n, \frac{108}{\Delta^2}\log\left(\frac{4 \cdot 108}{\Delta^2\alpha}\right)\right\}.$$

To see this, suppose first that $n \geq \beta$ where

$$\beta = \frac{108}{\Delta^2}\log\left(\frac{4 \cdot 108}{\Delta^2\alpha}\right).$$

Then, at $t = 2n$, the right hand side of (23) is

$$\frac{6}{\Delta}\sqrt{2n\log(2n/\alpha)} \leq n,$$

where the final inequality holds for all $n \geq \beta$, which was assumed. Now suppose that $n < \beta$, so that (23) should hold for $t \geq n + \beta$. Since the left hand side of (23) grows faster than the right hand side, it suffices to show that it holds at $t = n + \beta$. To this end, write

$$\frac{6}{\Delta}\sqrt{t\log(2t/\alpha)}\bigg|_{t=n+\beta} \leq \frac{6}{\Delta}\sqrt{(n+\beta)\log(2n/\alpha + 2\beta/\alpha)}$$

$$\leq \frac{6}{\Delta}\sqrt{2\beta\log(4\beta/\alpha)}$$

$$= \frac{72}{\Delta^2}\sqrt{\log\left(\frac{4 \cdot 108}{\Delta^2\alpha}\right)\log\left(\frac{4 \cdot 108}{\Delta^2\alpha}\log\left(\frac{4 \cdot 108}{\Delta^2\alpha}\right)\right)}$$

$$= \frac{72}{\Delta^2}\sqrt{\log\left(\frac{4 \cdot 108}{\Delta^2\alpha}\right)\log\left(\frac{4 \cdot 108}{\Delta^2\alpha}\right) + \log\log\left(\frac{4 \cdot 108}{\Delta^2\alpha}\right)}$$

$$\leq \frac{108}{\Delta^2}\log\left(\frac{4 \cdot 108}{\Delta^2\alpha}\right) = \beta,$$

where the final inequality uses the (loose) bound $\log\log(x) \leq \log^2(x)$.

### C.4   Proof of Proposition 4

First let us note that $(\mathcal{K}_t^Q)$ are indeed nonnegative supermartingales. Under the null $H_0'$, we have $\mathbb{E}_{H_0'}[\widehat{Y}_t^0 - \widehat{Y}_t^1] \leq \epsilon$, so

$$\mathbb{E}_{H_0'}[Q_t^R|\mathcal{F}_{t-1}] = 1 + \lambda_t(\mathbb{E}_{H_0'}[\widehat{Y}_t^0 - \widehat{Y}_t^1] - \epsilon) \leq 1.$$

Moreover, for $\lambda_t \in [\frac{-1}{1-\epsilon}, \frac{1}{1+\epsilon}]$, $Q_t \geq 0$. The argument is similar for $R_t$. Therefore, by Ville's inequality,

$$P(\exists t \geq 1 : \mathcal{K}_t^Q \geq 2/\alpha) \leq \alpha/2, \text{ and } P(\exists t \geq 1 : \mathcal{K}_t^R \geq 2/\alpha) \leq \alpha/2,$$

so the union bound gives $P(\exists t \geq 1 : \mathcal{K}_t^Q \geq 2/\alpha \text{ or } \mathcal{K}_t^R \geq 2/\alpha) \leq \alpha$. This implies that the proposed test, which involves rejecting if $\max\{\mathcal{K}_t^Q, \mathcal{K}_t^R\} \geq 2/\alpha$ is a level-$\alpha$ sequential test.

Now, note that ONS restricts $\lambda_t$ to be in $[-1/2, 1/2]$, which is a subset of $[\frac{-1}{1-\epsilon}, \frac{1}{1+\epsilon}]$. We can therefore use a similar ONS analysis as above, without needing to modify the guarantees to account for the range of $\lambda$. Recall from Section C.1 the definitions $g_t$, $S_t$, and $V_t$. Because $Q_t$ and $R_t$ have an extra $\epsilon$ term, we need to modify these terms for this new setting. Define

$$S'_t = \sum_{i=1}^{t}(g_i - \epsilon), \qquad V'_t = \sum_{i=1}^{t}(g_i - \epsilon)^2.$$

The guarantee given by (20) translates, in this setting, to

$$\mathcal{K}_t^Q \geq \frac{1}{V'_t}\exp\left\{\frac{(S'_t)^2}{4(V'_t + |S'_t|)}\right\},$$

where we've simply replaced $S_t$ and $V_t$ by $S'_t$ and $V'_t$. The same guarantee holds for $\mathcal{K}_t^Q$ (there $g_i$ is replaced with $-g_i$ but it makes no difference).

Results for asymptotic power (for both $\mathcal{K}_t^Q$ and $\mathcal{K}_t^R$) follow from similar arguments as in Section C.1 but replacing $S_t$ and $V_t$ with $S'_t$ and $V'_t$. Therefore, let us turn our attention to expected stopping time. We will consider the expected stopping time of the test based on $\mathcal{K}_t^Q$. Since identical arguments hold for $\mathcal{K}_t^R$, one need only multiply the result by 2 to get the overall expected stopping time.

Following what was done in previous proofs, write $\mathbb{E}[\tau] = \sum_{t=1}^{\infty} P(E_t)$, where $E_t = \{\log \mathcal{K}_t^Q < \log(1/\alpha)\}$. We have

$$\begin{aligned}
E_t &\subset \{(S'_t)^2 < 4(V'_t + |S'_t|)(\log(1/\alpha) - \log(1/V'_t))\}\\
&\subset \{(S'_t)^2 < 4\left(V_t + t\epsilon^2 + |S_t| + t\epsilon\right)(\log(1/\alpha) - \log(1/V'_t))\}\\
&\subset \left\{(S'_t)^2 < 4\left(V_t + t\epsilon^2 + \sum_{i\leq t}|g_i| + t\epsilon\right)(\log(1/\alpha) - \log(1/V'_t))\right\}.
\end{aligned}$$

Here we've used that $V'_t \leq V_t + t\epsilon^2$ and $|S'_t| \leq |S_t| + t\epsilon$. As in Section C.1, we have $W_t = \sum_{i\leq t}|g_i| \leq 2t$ and $V_t \leq 2t$ with probability $1 - t^2$ for all $t \geq 17$. Let $H_t$ be the event that both $V_t \leq 2t$ and $W_t \leq 2t$. Note that by the reverse triangle inequality, $|S'_t| \geq |S_t| - t\epsilon$. Therefore,

$$\begin{aligned}
E_t \cap H_t &\subset \{|S'_t|^2 < (16t + t(\epsilon + \epsilon^2))(\log(1/\alpha) + \log(2t + t\epsilon^2))\}\\
&\subset \{|S'_t| < \sqrt{(16t + t(\epsilon + \epsilon^2))(\log(1/\alpha) + \log(2t + t\epsilon^2))}\}\\
&\subset \{|S_t| < t\epsilon + \sqrt{(16t + t(\epsilon + \epsilon^2))(\log(1/\alpha) + \log(2t + t\epsilon^2))}\}\\
&\subset \{|S_t| < t\epsilon + \sqrt{18t(\log(1/\alpha) + \log(3t))}\},
\end{aligned}$$

where we've used that $\epsilon < 1$. With probability $1 - 1/t^2$, $|S_t| \geq t\Delta - \sqrt{2t\log(2t)}$. The minimum $t = T$ such that $t\Delta - \sqrt{2t\log(2t)} \geq t\epsilon + \sqrt{18t(\log(1/\alpha) + \log(3t))}$ is on the order of

$$T \lesssim \frac{1}{(\Delta - \epsilon)^2}\log\left(\frac{1}{\alpha(\Delta - \epsilon)^2}\right).$$

Therefore,

$$\mathbb{E}[\tau] = \sum_{t=1}^{\infty}P(E_t) \leq T + \sum_{t>T}(P(E_t \cap H_t) + P(H_t^c)) \leq T + \sum_{t>T}\frac{3}{t^2} \lesssim T.$$

Note that this is under the alternative, so $\Delta > \epsilon$ and $0 < \Delta - \epsilon < \Delta$.

# D  Simulation Details

Code to recreate all plots and run the simulations is available at `https://github.com/bchugg/auditing-fairness`. Here we provide more extensive details on each figure.

**Figure 1.** Given $\Delta$, we generate the two means $\mu_0$ and $\mu_1$ as $\mu_0 = 0.5 + \Delta/2$ and $\mu_1 = 0.5 - \Delta/2$. We take $\varphi(X)|\xi_b$ to be distributed as $\text{Ber}(\mu_b)$. (Thus, this simulates a scenario for which we witness the classifcation decisions, not e.g., a risk score.) We set $\alpha = 0.01$, so we reject when the wealth process is at least 100. We receive a pair of observations each timestep. Each experiment was run 100 times to generate the plotted standard deviation around the mean of each wealth process.

**Figure 2.** As above, we take the distribution of model observations $\varphi_t(X)|\xi_b$ to be $\text{Ber}(\mu_b(t))$. For the left hand side of Figure 2 we take $\mu_0(t) = \mu_1(t) = 0.3$ for $t = 1, \dots, 99$. At $t = 100$, we add a logistic curve to $\mu_1$. In particular, we let

$$\mu_1(t) = 0.3 + \frac{0.5}{1 + \exp((250 - t)/25)}, \quad t \geq 100.$$

We keep $\mu_0(t)$ at 0.3 for all $t$. For the right hand side of Figure 2, we let both $\mu_1$ and $\mu_0$ be noisy sine functions with different wavelengths. We let

$$\mu_0(t) = \frac{\sin(t/40)}{10} + 0.4 + \epsilon_t^0,$$

for all $t$, where $\epsilon_t^0 \sim N(0, 0.01)$. Meanwhile,

$$\mu_1(t) = \frac{\sin(t/20)}{10} + 0.4 + \frac{t}{1000} + \epsilon_t^1,$$

where, again, $\epsilon_t^1 \sim N(0, 0.01)$. The mean $\mu_1(t)$ thus has a constant upward drift over time. As before, we assume we receive a pair of observations at each timestep and we take $\alpha = 0.01$. We generate the means once, but run the sequential test 100 times in order to plot the standard deviation around the mean of the wealth process.

**Figures 3 and 4.** For a given sequential test and a given value of $\alpha$, we run (i) the test under the null hypothesis (i.e., with a fair model; we describe how we generated a fair model below), and (ii) the test under the alternative. Repeating 300 times and taking the average gives the FPR and average rejection time for this value of $\alpha$. This procedure is how the leftmost two columns of Figure 3 were constructed. The final column then simply plots the FPR versus the value of $\alpha$.

We used a vanilla random forest for both the credit default dataset and the US census data. For the credit default dataset, the model does not satisfy equality of opportunity [39] when $Y$ indicates whether an individual has defaulted on their loan, and $A$ indicates whether or not they have any university-level education. One can imagine loans being given or withheld on the basis of whether they are predicted to be returned; we might wish that this prediction does not hinge on educational attainment. For the census data, the model does not satisfy equality of opportunity when $A$ indicates whether an individual has an optical issues, and $Y$ indicates whether they are covered by public insurance. Admittedly, this example is somewhat less normative than the other. It is unclear whether we should expect perfect equality of opportunity in this context. However, we emphasize that our experiments are for illustrative purposes only. They are not meant as comments on the actual fairness or unfairness of these datasets. We interface with the census data by means the folktables package [51].

For the credit default dataset and random forest classifier, we have $\Delta = |\mu_0 - \mu_1| = 0.034$. We have $\Delta = 0.09$ for the census data. To construct the fair model (in order to test the null), we add $\Delta$ to the model predictions of the group with the lower mean. Thus, the distributions of predictions are different but the means are identical.

Figure 4 follows similar experimental logic, but we begin with a fair model (i.e., group predictions with the same mean, generated as above), and then switch to the unfair random forest classifier at time $t = 400$.

**Figure 5.** We use US health insurance data [52], which is synthetic data generated based on US census information. We train a logistic regression model to predict the risk of non-repayment based on 11 covariates which include age, BMI, gender, region, and whether the individual is a smoker. We find that the predicted risks are different among males (0.235) and females (0.174), so we use $A$ as gender and use equality of opportunity as our notion of group fairness.

We construct three data collection policies, $\pi_1, \pi_2$, and $\pi_3$, which are probability distributions over four regions: NE, SE, NW, SW. Data-collection works by first sampling a region with the probability prescribed by $\pi_i$, and then sampling an individual in that region uniformly at random. We defined the three strategies as follows:

$$\pi_1(NE) = 0.1, \ \pi_1(NW) = 0.2, \ \pi_1(SE), \ 0.3, \ \pi_1(SW) = 0.4,$$
$$\pi_2(NE) = 0.05, \ \pi_2(NW) = 0.15, \ \pi_2(SE), \ 0.25, \ \pi_2(SW) = 0.55,$$
$$\pi_3(NE) = 0.05, \ \pi_3(NW) = 0.1, \ \pi_3(SE), \ 0.15, \ \pi_3(SW) = 0.7.$$

We also ran our-betting based method with data sampled uniformly for the population, which served as the comparison point. Of course, as is expected from Proposition 1 and 2, data sampled uniformly from the population yields better results. We also compared to permutation tests run on data sampled uniformly from the population. These were implemented as described above and we do not belabor the details here.

