# OpenReview forum: "Auditing Fairness by Betting"
_NeurIPS.cc/2023/Conference — NeurIPS 2023 spotlight_

### Official Review · Reviewer_YGbH · 2023-07-05

**Soundness:** 4 excellent
**Presentation:** 4 excellent
**Contribution:** 4 excellent
**Rating:** 8
**Confidence:** 4

**Summary:**

The paper proposes a safe anytime-valid inference (SAVI) testing procedure to audit the potential unfairness of a classifier or regression model. The paper posits group-conditioned fairness as a test of equality of means under some finite set of conditions, this definition includes several common fairness notions such as equality of odds and equality of opportunity. In this setting, the authors propose a technique allows for sequential testing of equality of means that can accommodate repeated testing, finite or infinite testing time horizons, batched testing, and importance weighted testing. The authors also provide expected stopping times for their testing procedure.

**Strengths:**

The paper is well written and structured. All the necessary information to understand the paper's contributions is provided within the main text. Several practical considerations (time horizon, batched testing, importance weighting) are clearly outlined and addressed. The method itself is extremely simple to implement.

**Weaknesses:**

Overall the paper is very solid. The main weakness I see is that there is no immediate way of extending these ideas to settings where equality of means may be too strict of a definition, and instead one would want to test that the absolute difference of means falls below some epsilon value.

**Questions:**

See weaknesses above.

**Limitations:**

Yes

---

> ### Author Rebuttal · Authors · 2023-08-09
>
> We thank the reviewer for their interest in the paper!
>
> We agree that exact equality of means may be too precise a condition in practice (though we note that most other group fairness papers work with these definitions). It is in fact possible to handle a null of the form $H_0: |\mu_1-\mu_0|<\epsilon$. The idea is to use two supermartingales, one to detect if $\mu_1\geq \mu_0 +\epsilon$ and one to detect if $\mu_0\geq \mu_1 + \epsilon$. We can then examine the maximum of these two processes over time, rejecting the null if either one exceeds $1/\alpha$. Unfortunately, we were unable to discuss this in the submitted paper due to space constraints but, if the paper is accepted, we plan on using the additional page to explain this idea in more detail.

---

> > ### Comment · Reviewer_YGbH · 2023-08-16
> >
> > Thank you for the explanation. That addresses my sole concern and I do thank the authors for a very interesting read. Great work!

---

### Official Review · Reviewer_T7v6 · 2023-07-06

**Soundness:** 3 good
**Presentation:** 4 excellent
**Contribution:** 3 good
**Rating:** 6
**Confidence:** 4

**Summary:**

This work introduces practical and efficient methods for auditing the fairness of deployed classification and regression models. The methods are sequential, allowing for continuous monitoring of incoming data, and can handle distribution shifts resulting from changes in the model or the underlying population. The approach is based on recent advancements in anytime-valid inference and game-theoretic statistics. The methods are demonstrated to be effective through experiments on benchmark fairness datasets.

**Strengths:**

Originality:

The concept of developing a system that can continuously monitor and track the fairness of deployed real-world systems in a sequential evaluation context is a novel contribution. The idea of extending fairness evaluation beyond the development phase is an innovative approach.

Quality:

The paper demonstrates a high level of quality, with well-founded theoretical reasoning and rigorous analysis. The authors present a solid theoretical foundation that supports their proposed methods.

Clarity:

The paper is written in a clear and understandable manner. The research problem is effectively formulated, and the theoretical foundations are explained comprehensively. The clarity of the paper contributes to its accessibility and readability.

Significance:

Ensuring fairness in systems goes beyond the initial testing phase and requires continuous monitoring of deployed models. The significance of the paper lies in its contribution to the field by providing a way to quantify and evaluate the fairness of deployed models. This capability is becoming increasingly important as the need for ongoing monitoring and accountability in real-world systems grows. The paper's focus on continuous fairness monitoring and its impact on the research community make it a valuable contribution to the field.



**Weaknesses:**

It would greatly enhance the paper's findings to include more than two datasets for result verification. Additionally, since only one model (Random Forest) was trained on the dataset, it would be intriguing to explore the effects of different fairness methods, such as in-processing, post-processing, and preprocessing approaches, by testing multiple models. By incorporating these suggestions, the paper's results would become more robust and comprehensive, providing a more thorough understanding of the proposed methods and their performance across various datasets and models.

**Questions:**

I would like to discuss a thought experiment regarding a fair model that exhibits some degree of non-determinism. In this scenario, the model generally produces fair outputs for a given instance, but with a certain probability (e.g. 10%) the same instance would have an unfair output. My intention in raising this thought is to gain insights into the robustness and reliability of your framework. I am curious to understand whether such non-deterministic behavior would impact or challenge the effectiveness of your proposed framework.

**Limitations:**

I acknowledge that cost considerations and reaching a sufficient amount of evidence to make a decision are factors that may lead to the discontinuation of data collection. However, it is crucial to recognize that the fairness of a deployed system can evolve over time, necessitating continuous evaluation to ensure ongoing fairness. In addition, the emergence of regulatory frameworks like the AI Act in the EU emphasizes the importance of continuous evaluation, as it may become a legal requirement. These factors highlight the need for continuous monitoring and evaluation to address fairness concerns and comply with legal obligations. So even if the null is rejected at some other time in the future this could not be seen any more. How does the framework could handle this? Is this a limitation?

---

> ### Author Rebuttal · Authors · 2023-08-09
>
> We thank the reviewer for posing several thoughtful questions.
>
> Regarding the weaknesses, Section 4 now includes several new experiments that employ various data collection policies on a new dataset (an insurance dataset) and with a new model (logit).  We test three different data collection policies, all based on the location of the individual, which is one of NE, NW, SE, SW. In the original dataset, individuals are roughly evenly split among these locations. We test three policies, each of which places successively more weight on the SW region. We refer the reviewer to the attached pdf for figures of the results. Overall, we see that the method continues to perform well under all three policies, though it requires more steps to reject than if the streams are representative of the population. This is as suggested by comparing Proposition 1 and 2).
>
> Regarding the reviewer's question: yes, our method allows for potentially random behavior in the model. Note that our method tests standard definitions of group fairness, which require that the expected performance of the model (on some metric) is equal between groups. According to these definitions, a model which is fair 90\% of the time and unfair 10\% of the time is ``unfair": the 10\% unfair behavior means that the average performance is no longer identical between groups. Our method uses samples of (features, labels, predictions) to test whether the expected performances are equal, and the distribution of these samples can include any internal randomness of the model as well as the distribution over features and labels.
>
> Regarding limitations: Suppose the null is initially rejected at time $t_1$. If the system changes substantially in response to this original audit, then we may simply begin a new auditing process afresh (if the system did not change, we may unsurprisingly simply reject the same null again). However, some sort of correction is required to perform multiple audits. For instance, we might run the second audit with $\alpha/2$, the third with $\alpha/4$, and so on in order to ensure level-$2\alpha$ across all audits. Even better, we might consider running an online false discovery rate controlling procedure (several of these have been suggested in the recent ML and Statistics literatures, we will point to them in our paper).

---

> > ### Comment · Reviewer_T7v6 · 2023-08-14
> >
> > Thank you for your response and clarification.
> >
> > I think the new experiments and the clarification about the limits are valuable and a nice extension of the work.
> >
> > About my thought experiment: I think I did not formalize it in a clear way, or even the thought was not a good support for my point. The randomness I was talking about was not between groups, but more like on instances. So say you run the model multiple times with the same instance than 90% of the time the model will output the class 1 and 10% of the time the class 0. So its more on the individual fairness side. I see that your method is built for group fairness but I was wondering if you can extend it to individual fairness?

---

> > > ### Author Response · Authors · 2023-08-14
> > >
> > > We also think extensions to the individual fairness (IF) setting are worth considering. Unfortunately, it's difficult to see at first glance how to make our particular approach work for IF because it doesn't rely on distributional differences (e.g., equality of means). Instead, IF only requires ``similar'' treatment among people who have similar features. Thus, there isn't typically a notion of repeated sampling from the same distribution (since each individual has different features), which is fundamental to our strategy.
> > >
> > > Of course, if there is some notion of repeated sampling (as in your thought experiment) then our method might be applied. For instance, there has been some recent work on IF in clusters, i.e., people in the same cluster should be treated equally. In this case, we could apply our method using each cluster as the distribution, where our hypothesis would be that the average difference between the predictions in that cluster is zero. We would then play $k$ games simultaneously, where $k$ is the number of clusters.
> > >
> > > We hope this helps answer the reviewer's excellent question!

---

> > > > ### Comment · Reviewer_T7v6 · 2023-08-15
> > > >
> > > > Thank you for your response. I would say that adding this idea about IF and clustering could be a nice extension to the paper and could even be placed in the conclusion section.

---

### Official Review · Reviewer_sAy5 · 2023-07-06

**Soundness:** 4 excellent
**Presentation:** 4 excellent
**Contribution:** 4 excellent
**Rating:** 8
**Confidence:** 4

**Summary:**

This work proposes an approach based on recent results of super-martingales to detect when a certain predictor is demographically unfair. Studying different formulations of group fairness (such as equal opportunity, demographic parity, etc), the authors frame the problem as that of sequential hypothesis test, where a null corresponding to equalized metrics across groups. The problem is then to reject the null, if the alternative is true, as soon as possible while controlling for type-1 error. Because this is set up as a sequential hypothesis test, this implies strong conditions on the probability of false positive (i.e. for any future time). Relying on the idea of “betting”, the authors then design a pay-off function that allows the “payoff” to be a super-martingale under the null (and remain controlled), whereas the payoff grows exponentially under the alternative. While the authors derive their methodology a a “basic”/standard, they further comment on how the same approach can be minimally modified to accommodate for data collected at different rates from different demographic groups, with different data-collection policies (not iid), and under “distribution shifts”. They finally showcase their method in a couple of real world datasets, comparing with relatively simple alternative.

**Strengths:**

- The studied problem is quite important and timely, given the increasing use of predictive algorithms in society.
- The proposed solution is relatively simple and elegant, provably correct, and seems effective in practice.
- This paper is beautifully written, and it was a pleasure to read.
- The experiment results are very nice and easy to understand

**Weaknesses:**

- The experiment results do not seem to address the time-varying data collection policies setting.


**Questions:**

1. A potentially trivial question is how to employ the same ideas to not only equal opportunity but also to equalized odds (which requires not only equal TPR but also FPRs to be the same). Is it true that in this case one can simply generalize the null hypothesis as one where both rates being the same, and the results would naturally follow? A footnote on this might be useful.

2. While their approach seems to work quite nicely in practice, I wonder about the tightness of Proposition 1 (and its corollaries that follow). What are the typical constants associated with this bound? Moreover, do the authors think it would be possible to obtain a lower bound on $\mathbb E[\tau]$ as a function of $\Delta$?

3. As far as I understand, the experiments section did not showcase their results on being able to incorporate different data-collection policies (Sec 3.2). Could the authors comment on why this is the case? Even if the data they used involved static policies, could they simulate different policies a posteriori to showcase their results?

**Limitations:**

The results all hinge on the auditing process being conducted properly. The paper realizes this and appropriately mentions it in the limitations section.

---

> ### Author Rebuttal · Authors · 2023-08-09
>
> We thank the reviewer for their enthusiasm regarding the work! To address the main weakness, we have added experiments for various data collection policies. We discuss the details in point 3 below, and respond to the other questions in order:
>
>
> 1. We can handle equalized odds by playing two games simultaneously: One would check whether the TPRs are the same and the other whether the FPRs are the same. We thank the reviewer for raising this point; we will make sure to add a remark about it.
> 2. The asymptotic notation in Proposition 1 is hiding a constant of 81. However, we did not attempt to optimize the constant; doing so could perhaps shave off a factor of 2 or more. It is also possible, using information-theoretic arguments, to get a lower bound over all sequential tests of the form $\mathbb{E}[\tau]\gtrsim\log(1/\alpha)/\Delta^2$. We will add a discussion of this fact to the text.  Notice that, up to constants, our upper bound is off of this lower bound by a log-factor. An interesting open question is whether this log-factor difference between upper and lower bounds can be removed, either through a tighter lower bound (perhaps specific to our test), or a tighter upper bound.
> 3. Section 4 now includes several experiments that employ various data collection policies. We also use a new dataset (an insurance dataset) and a different model (logit) in order to further test our methods in new settings. We test three different data collection policies, all based on the location of the individual, which is one of NE, NW, SE, SW. In the original dataset, individuals are roughly evenly split among these locations. We test three policies, each of which places successively more weight on the SW region. We refer the reviewer to the attached pdf for figures of the results. Overall, we see that the method continues to perform well under all three policies, though it requires more steps to reject (as suggested by comparing Proposition 1 and 2).

---

> > ### Comment · Reviewer_sAy5 · 2023-08-12
> > **Questions and concerns addressed**
> >
> > I thank the authors for addressing my questions and comments.

---

### Official Review · Reviewer_am4v · 2023-07-07

**Soundness:** 4 excellent
**Presentation:** 3 good
**Contribution:** 3 good
**Rating:** 7
**Confidence:** 3

**Summary:**

This paper clarifies the issue of fairness auditing, wherein an auditor verifies the fairness of a system through sequential interaction with it. The goal is to develop a methodology that facilitates anytime-valid tests of system fairness. Essentially, this involves devising a sequence of test statistics for the null hypothesis (i.e., the system is fair), enabling the auditor to conduct statistical tests using these statistics at any given stopping time. To achieve this, the authors design a payoff for the invalidity of observed information against the null hypothesis. A sufficiently high value of this payoff can safely lead to the rejection of the null hypothesis. Both theoretical and experimental analyses of the proposed method substantiate the efficacy of this sequential fairness testing approach.

**Strengths:**

This is a well-written and fascinating paper. Recently, the problem of fairness auditing has garnered attention, driven by societal demands for fair systems incorporating machine learning techniques. Sequential testing of fairness has become an essential tool for such audits. This paper delivers a robust technical solution for sequential fairness testing, offering impressive theoretical and empirical analyses of its performance.

The principal technical innovation is the reduction of the fairness auditing problem into a two-sample testing problem. This transformation facilitates the application of the established and theoretically sound methodology of anytime-valid two-sample testing.

The empirical experiments conducted in this paper effectively illustrate the potential for achieving higher power with fewer interactions, all while controlling the Type-I error in comparison to the permutation test. This evidence underscores the practicality of the proposed method.

**Weaknesses:**

A notable weakness is the limited innovation in methodology. The construction of the testing algorithm is merely an application of the anytime-valid two-sample testing developed by Shekhar and Ramdas. Additionally, even though Shekhar and Ramdas did not account for adjustments to time-varying data collection and distributional shifts, such an extension might be relatively straightforward.

In the context of time-varying data collection, the authors make the assumption that the density of the covariate distribution is readily accessible. This assumption, however, isn't practically feasible. Therefore, it would be advantageous to address the limitations of this assumption more thoroughly.

**Questions:**

The existence of group-wise streams may not be feasible in practice. In a typical scenario, an auditor might receive a sequence of i.i.d. data drawn from a composite distribution that combines the group-wise distributions. Therefore, it would be more beneficial to illustrate the application in a situation where a single, mixed stream is present.

**Limitations:**

There are no specific limitations or potential negative social impacts that need to be addressed in relation to this paper.

---

> ### Author Rebuttal · Authors · 2023-08-09
>
> We thank the reviewer for their kind words on the paper.
>
> Regarding time-varying data collection, we agree that the density, $\rho_t$, will not always be known. However, we had in mind situations in which knowing the density is feasible.  Consider, for instance, algorithms deployed across a database of users (e.g., social media, e-commerce). Precise population characteristics might be known by virtue of the fact that users must sign up for the application and provide various information. Another example comes from considering data-collection policies which use demographic features, income bracket, or general socioeconomic indicators. In this case the population level quantities may be known via state and federal statistics.
>
> That being said, we recognize that the reviewer is correct and that we may not always have access to the density. We have therefore added an ``extensions" section which discusses how our method can be amended when only an estimate of the density is available, in addition to bounds on its multiplicative error. The idea is straightforward: we reweight the terms $\omega_t^b\hat{Y}_t^b$ in the payoff function by these bounds. The resulting process can be shown to be a *super*martingale (as opposed to a martingale), and therefore results in a more conservative test in practice. The mathematical guarantees, however, still hold.   We thank the reviewer for raising this interesting question.
>
> Finally, we also agree with the reviewer regarding group wise streams. Section 3.2 in the submitted paper discusses how to reformulate our methods to handle a single stream of data. The idea is to wait to bet until there are new predictions available for both groups.
> If multiple predictions for one group have accrued in the interim, we then bet with their average.

---

> > ### Comment · Reviewer_am4v · 2023-08-13
> >
> > Thank you for the response from the authors. I wish to maintain my initial score.
> >
> > Could the authors address the first weakness I highlighted? To me, the connection between the fair audit problem and the two-sample testing problem is captivating and feels novel. However, I can understand why someone might view the reduction as obvious, even if no one has expressed this concern, and consider the proposed method merely an application of Shekhar and Ramdas. I believe the paper would benefit from a more in-depth discussion differentiating the proposed method from the work of Shekhar and Ramdas.
> >
> > Regarding the single stream, I would appreciate it if the paper, rather than merely showcasing a methodology for handling a single stream, could provide an analysis of the stopping time within that single stream context.

---

> > > ### Author Response · Authors · 2023-08-14
> > >
> > > We understand the reviewer's concern. The paper is indeed an application of ideas in Shekhar and Ramdas---and game-theoretic statistics more generally---to the fairness setting, but we believe our paper makes additional contributions on top of previous work.
> > >
> > > First, Shekhar and Ramdas work in a fairly abstract setting, using betting strategies based on kernels and oracle tests. Consequently, the guarantees on their test are more complicated and analyzes the expected stopping time in terms of a regret sequence, properties of the kernel, and various quantities that are difficult to calculate in practice. (See Theorem 1 in their paper.) Our bound, on the other hand, is stated in terms of basic quantities such as the distance between the two means. The analysis is also simplified, not requiring any regret analysis or kernel properties. We believe this more straightforward bound and analysis makes the work easier to comprehend for practitioners.
> > >
> > > Second, we have extended the method and analysis to handle both randomized data collection policies and distribution drift, both of which we think are ubiquitous features of fairness testing in practice. Again, our bounds in these settings are straightforward and easily understood by practitioners.
> > >
> > > Third, at the request of another reviewer, we have added a section on how to deal with composite nulls of the form $H_0: |\mu_0-\mu_1|<\epsilon$, for some $\epsilon>0$, which further extends the applicability of our framework. The idea is to use two supermartingales, one to test whether $\mu_0\geq \mu_1 + \epsilon$ and one to test whether $\mu_1\geq \mu_0 + \epsilon$.
> > > As that reviewer pointed out, it might be too much to ask of a system to have the means be precisely equal.
> > >
> > > We hope the reviewer agrees that these differences from Shekhar and Ramdas make the paper a valuable contribution. We will be sure to highlight these differences more thoroughly in the final version.
> > >
> > > Regarding the streams, we agree that an explicit proposition tailored to this setting is valuable for the reader.
> > > In the appendix we've added a theorem which analyzes the expected stopping time in terms of the number of ``switches'' in the single stream (in addition to the total number of observations and other quantities), i.e., the number of times that we receive some number of decisions from one group followed by a decision from the other group.

---

### Author Rebuttal · Authors · 2023-08-09

Thanks to all reviewers for their thoughtful questions and comments!

---

### Decision · Program_Chairs · 2023-09-21

**Decision:**

Accept (spotlight)

**Comment:**

All reviewers agreed that the paper should be accepted, with some reviewers describing the paper as "fascinating", "simple and elegant", "important and timely" and "a pleasure to read". Some improvements that the reviewers would like to see are a clarification of the relationship of this work to Shekhar and Ramdas, and an extension of the hypothesis test to handle a null in which the means are not exactly equal.